# Induction of Glucoraphasatin Biosynthesis Genes by MYB29 in Radish (*Raphanus sativus* L.) Roots

**DOI:** 10.3390/ijms21165721

**Published:** 2020-08-10

**Authors:** Ji-Nam Kang, So Youn Won, Mi-Suk Seo, Jeongyeo Lee, Si Myung Lee, Soo-Jin Kwon, Jung Sun Kim

**Affiliations:** 1Genomics Division, National Institute of Agricultural Sciences, Rural Development Administration, Jeonju 54874, Korea; greatnami@korea.kr (J.-N.K.); soyounwon@korea.kr (S.Y.W.); sms1030@korea.kr (M.-S.S.); tataby@korea.kr (S.M.L.); 2Plant Systems Engineering Research Center, KRIBB, Daejeon 34141, Korea; leejy@kribb.re.kr

**Keywords:** glucoraphasatin, glucosinolates, MYB transcription factors, radish transcriptome, GRS1

## Abstract

Glucoraphasatin (GRH) is a specific aliphatic glucosinolate (GSL) that is only abundant in radish (*Raphanus sativus* L.). The gene expression regulating GRH biosynthesis in radish is still poorly understood. We employed a total of 59 radish accessions to analyze GSL profiles and showed that GRH was specific and predominant among the aliphatic GSLs in radish roots. We selected five accessions roots with high, moderate and low GSL biosynthesis, respectively, to conduct a comparative transcriptome analysis and the qRT-PCR of the biosynthesis genes for aliphatic GSLs. In this study, among all the accessions tested, roots with the accession RA157-74 had a high GRH content and showed a significant expression of the aliphatic GSL biosynthesis genes. We defined the genes involved in the GRH biosynthesis process and found that they were regulated by a transcription factor (*RSG00789*) at the MYB29 locus in radish roots. We found 13 aliphatic GSL biosynthesis genes regulated by the *RSG00789* gene in the GRH biosynthesis pathway.

## 1. Introduction

Glucosinolates (GSLs) are secondary metabolites containing nitrogen and sulfur, mainly found in plants. Approximately 200 GSLs are known to exist naturally in plants [1]. GSLs are classified into several groups by the structure of the side chain [2]. According to the modified amino acid content, three main groups exist: aliphatic, indolic and aromatic. Aliphatic GSLs are derived from methionine, while indolic and aromatic GSLs originate from tryptophan and phenylalanine, respectively [3].

Aliphatic GSL biosynthesis in the Brassicaceae family is well understood. It involves three independent phases: (i) chain elongation of precursor amino acids, (ii) core structure synthesis by partial amino acid conversion, and (iii) chain modification of the side amino acids [1]. Major genes involved in GSL biosynthesis have been reported in the Brassicaceae family, including in *Arabidopsis thaliana*, two *Brassica rapa* (ssp. *pekinensis* and ssp. *chinensis*), three *Brassica oleracea* (var. *capitata*, var. *italica* and var. *alboglabra*) and *Raphanus sativus* [2,4,5,6,7,8,9,10,11,12]. Major enzymes and transporters for aliphatic GSL biosynthesis in the Brassicaceae family include branched-chain aminotransferases (BCATs), methylthioalkylmalate synthases (MAMs), isopropyl malate isomerases (IPMIs), isopropylmalate dehydrogenase (IPMDH), bile acid transporter 5 (BAT5), CYP79 cytochrome P450 monooxygenases (CYP79F1), CYP83 cytochrome P450 monooxygenase (CYP83A1), S-alkyl-thiohydroximate lyase (SUR1), UDP-glucosyl transferase 74 (UGT74), sulfotransferases (SOTs) and flavin-monooxygenase glucosinolate S-oxygenases (FMO_GS-OXs_) [9,10,11,12]. MYB28 and MYB29 are transcription factors that regulate the genes involved in aliphatic GSL biosynthesis [6].

The radish (*Raphanus sativus* L.) is an economically important crop belonging to the Brassicaceae family, which also includes cabbage, kale and broccoli [13]. The radish has been used for high nutritional and medicinal foods. In particular, the taproot and young pods of radish are mainly used for human consumption in East Asia, including Korea, China and Japan [13,14]. The taproot has been known to contain minerals, vitamins fiber and GSLs, which are potentially important for human health [15].

The GSL composition of radish has been well established from several studies [16,17,18,19,20,21]. It contains a specific aliphatic GSL called glucoraphasatin (GRH, 4-methylthio-3-butenyl-GSL), which is only abundant in radish. Raphasatin is an isothiocyanate (ITC) derived from GRH and has an antitumorigenic effect that promotes anticancer activity. ITCs are a beneficial compound for human health, effecting a protection against carcinogenesis, inflammation and cardiovascular disease [18].

Although GRH is important as a biological and potential health promoting compound, GSL metabolism is mainly studied in broccoli, kale, cabbage and Chinese cabbage. Few transcriptome studies of GSL biosynthesis in radish have been reported, and they have not shown the molecular mechanism of GRH biosynthesis [11,12]. The comprehensive gene expression profile of GRH biosynthesis in radishes is still unknown.

In this study, taproots of 59 radish accessions were subjected to an examination of the GSL profile. GRH was significantly predominant when compared with other GSLs in radish roots. We verified the gene expressions involved in the GRH biosynthesis pathway with a comparative transcriptome analysis. The analysis of RNA sequencing and quantitative RT-PCR (qRT-PCR) showed that major genes involved in GRH biosynthesis were strongly regulated by MYB29, which was not dependent on the GRH content of the radish roots. We investigated the MYB inducible mechanism of GRH biosynthesis in radish roots and identified those accessions that had a high GRH biosynthesis. These results may be useful in order to breed valuable new radish cultivars because the GRH content is an important and major trait in radishes.

## 2. Results

### 2.1. GSL Profiles in Radish Roots

To examine the GSL profiles in radish roots, 166 individual radish plants, two or three biological repeats from a total of 59 accessions, were used as the genetic sources. Six different aliphatic GSLs (glucoraphanin (GRA), glucoraphenin (GRE), gluconapin (GNA), glucobrassicanapin (GBN), glucoerucin (GER) and glucoraphasatin (GRH)) and four different indole GSLs (4-hydroxyglucobrassicin (4HGBS), 4-methoxyglucobrassicin (4MOGBS), neoglucobrassicin (NGBS) and glucobrassicin (GBS)) were detected in the roots of the 59 radish accessions, along with two unknown GSLs (Table 1 and Appendix A). Detailed information including the chemical name and side chains is found in Refs. [8,19,22,23,24]. An abundance of GRH was well demonstrated in previous studies on radish roots [17,18]. As expected, GRH was predominant in most of the accessions. The median of GRH content among all the accessions was 39.7 µmol·g^−1^ dry weight (wt.), which accounts for 88.8% of the total GSLs. Meanwhile, the GRA, GRE, GNA, GBN, GER, 4HGBS, 4MOGBS, NGBS and GBS contents were very low or not detected (Figure 1 and Appendix A).

### 2.2. RNA Sequencing and Mapping of Radish Reference Genomes

Among the radish accessions, RA502-92 roots were the low-abundance GRH accessions (LGRHA), and RA157-74 roots were the high-abundance GRH accessions (HGRHA). The GRH level was on average 791 times higher in the HGRHA roots when compared with the LGRHA roots. We selected five accessions for a comparative analysis of GRH biosynthesis. IT119282-15, IT119238-8 and RA280-82 roots were analyzed as the moderate GSL biosynthesis samples (Table 2 and Figure 2).

RNA sequencing was performed with a total of 15 radish root samples (three biological repeats from each of the above five accessions). 25,046,852~59,463,280 reads were produced from each sample after quality trimming. The Q30 values were from 97.38–98.28%, and the overall GC percentages were 44.78–47.16%. The average mapping ratio was 68.31% (Appendix A).

### 2.3. Analysis of Differentially Expressed Genes between Roots of the Radish Accessions

To investigate the expression level of genes involved in GSL biosynthesis, transcriptome data from the LGRHA and HGRHA roots were used for the analysis of differentially expressed genes (DEGs). A total of 2597 genes were up-regulated in the HGRHA roots when compared with the LGRHA roots, while 2417, 1654 and 1916 genes were up-regulated in the HGRHA roots when compared with the IT119282-15, IT119238-8 and RA280-82 accessions, respectively (Appendix A).

Gene ontology (GO) enrichment analysis was performed to classify the predicted functions of DEGs that were up-regulated in the HGRHA when compared with the LGRHA roots. The DEGs were classified into three main GO groups, consisting of biological processes, cellular component, and molecular function, including abundant GO terms. The top five GO terms in the biological processes group were the hydrogen sulfide biosynthetic, glucosinolate biosynthetic, sulfur compound biosynthetic, glycosyl compound biosynthetic and amine biosynthetic processes. These highly linked GO terms were related to biosynthesis processes that stimulate biological responses (Figure 3 and Appendix A). Similar results were obtained when comparing the HGRHA roots with the IT119282-15, IT119238-8 and RA280-82 accession roots (Appendix A).

### 2.4. Identification and Expression Analysis of GSL Biosynthesis Genes in Radish Roots

We selected a total of 64 genes with paralogues involved in GSL biosynthesis by a functional annotation in radish roots (Appendix A), and the expression values (FPKM) of these genes are shown in the heat map in Figure 4B. Major genes involved in aliphatic GSL biosynthesis were highly expressed in the HGRHA roots (Figure 4B). 

MYB28 and MYB29 transcription factors are mainly defined as regulators for aliphatic GSL biosynthesis genes in Arabidopsis [6]. Three MYB28 and two MYB29 transcription factors were annotated (Appendix A). The *MYB29* gene (*RSG00789*) was significantly differentially expressed in the HGRHA roots only. Interestingly, no significant expression of three paralogous *MYB28 genes* (*RSG53581*, *RSG23384*, *RSG16088*) was observed in the HGRHA roots when compared with all the tested accessions, despite the fact that MYB28s are the major transcription factors for aliphatic GSL biosynthesis genes [25]. The *RSG53581* locus was positively expressed in the HGRHA roots (Figure 4B and Appendix A). Two *MYB34* paralogues (*RSG13843*, *RSG52510*) were identified in radish roots (Figure 4B and Appendix A), and they are reported to regulate the expression of indolic GSL biosynthesis genes [24].

Genes involved in the side chain elongation step that were highly expressed in the HGRHA roots when compared with the tested accessions included the *BCAT4* (*RSG13682*) encoding branched-chain aminotransferase 4, two loci of *MAM1* (*RSG13852* and *RSG24637*) encoding methylthioalkylmalate synthase 1, *IPMI_LSU1* (*RSG39553*) encoding isopropyl malate isomerase large subunit 1, *IPMI_SSU2* (*RSG05959*) encoding isopropyl malate isomerase 2, *IPMI_LSU1* (*RSG39553*) encoding isopropylmalate dehydrogenase 1 and *BAT5* (*RSG43366*) encoding bile acid transporter 5. Meanwhile, the *BCAT3* gene (*RSG13682*) encoding branched-chain aminotransferase 3 was similarly expressed in all radish accessions (Figure 4 and **Appendix A**). It is known that BCAT3 acts in a supplementary manner to BCAT4 for the transamination of methionine to 2-oxo acid [26].

*CYP79F1* (*RSG07720*) and *CYP83A1* (*RSG12342*) genes, belonging to the cytochrome p450 monooxygenase family and involved in the core structure synthesis step, were shown to have a high FPKM score in the HGRHA roots. SUR1 (*RSG38701*) encoding S-alkyl-thiohydroximate lyase 1, *UGT74B1* (*RSG35660*) and *UGT74C1* (*RSG13253*) encoding UDP-glucosyl transferases 74, and *SOT17* (*RSG03557*) and *SOT18* (*RSG07388*) encoding sulfotransferases showed a significantly high expression in the HGRHA roots. Other paralogues (*RSG12634*, *RSG34220*, *RSG12630*, *RSG12631*, *RSG12632*, *RSG12635*, *RSG32297*, *RSG32298*, *RSG49978*) encoding SOT18 were at basal expression levels in all accessions (Figure 4B and Appendix A). The expression of *CYP79A2* (*RSG02731*, *RSG08366*, *RSG38144*), *CYP79B2* (*RSG17908*, *RSG23431*, *RSG33798*), *CYP79B3* (*RSG34152*), *CYP83B1* (*RSG48186*) and *SOT16* (*RSG28254*, *RSG07387*) genes were at low levels of expression in the HGRHA roots when compared with all the tested accessions (Figure 4B and Appendix A). In Arabidopsis, these genes are involved in the biosynthesis of aromatic and indolic GSLs derived from phenylalanine or tryptophan [27].

In the Brassicaceae family, the side chain modification in aliphatic GSL biosynthesis is divided into 3, 4 and 5 carbon stages after the core structure formation, and then the side chains are modified by activating oxygenation, hydroxylation, alkenylation, benzoylation and methoxylation [28]. We verified six *FMO_GS-OX_* genes (*RSG51376*, *RSG57085*, *RSG57158*, *RSG15645*, *RSG14855*, *RSG27761*) encoding flavin-monooxygenease glucosinolate S-oxygenases and a *GS-OH* gene (*RSG00041*) belonging to the 2-oxoglutarate and Fe (II)-dependent oxygenase superfamily of proteins, which are responsible for the modification of the aliphatic GSLs [28]. Additionally, we found seven *CYP81F* genes (*RSG10180*, *RSG11506*, *RSG29212*, *RSG29215*, *RSG40296*, *RSG10178*, *RSG23271*) belonging to CYP81 cytochrome P450 monooxygenases and five *IGMT* genes (*RSG15203*, *RSG26550*, *RSG59991*, *RSG15204*, *RSG06937*) encoding *O*-methyltransferases for the modification of indolic GSLs. *AOP2* genes encoding 2-oxoglutarate-dependent dioxygenases were not detected in this study. However, the expression levels of these genes were very low in all the radish accessions (Figure 4 and Appendix A). It is known that the production of GRH is regulated by a single recessive gene in radish named glucoraphasatin synthase 1 (*GRS1*), which belongs to the 2-oxoglutarate and Fe (II)-dependent oxygenase superfamily [29,30]. The *GRS1* gene (*RSG02297*) was identified by functional annotation and was strongly expressed in the HGRHA roots (Table 3 and Figure 5A) when compared with all the tested accessions. It has been reported that the accumulation of GRH occurs via the conversion of GER, a process which is catalyzed by GRS1 (Figure 5B) [29,30].

We examined the expression of 17 DEGs using qRT-PCR. The oligo sequences used for qRT-PCR are listed in Appendix A, and *RsActin* was used as an internal control for the relative expression analysis. The qRT-PCR analysis strongly supported our RNA sequencing data (Figure 6).

### 2.5. GRH Content and Expression of GRH Biosynthesis Genes in Radish Roots

The expression of genes involved in GRH biosynthesis was significantly high in the HGRHA roots when compared with all the tested accessions (Figure 4, Figure 5 and Figure 6 and Appendix A). We would expect that the GRH content would be high in radish roots with a high expression of GRH biosynthesis genes. However, unexpected expression patterns were revealed among the radish accessions. For example, the GRH content was 274 times higher in the IT119238-8 accession roots when compared with the LGRHA roots (Table 2 and Figure 2). However, the expression levels of GRH biosynthesis genes were low in both accessions, with none being significantly differentially expressed. In fact, several genes were down-regulated in the IT119238-8 accession roots. There was a similar result between the LGRHA and RA280-82 accession roots (Figure 7 and Appendix A). RA166-77 accession roots had a high GRH content (98.42 µmol·g^−1^ dry wt.), which was not significantly different from the HGRHA roots (Figure 8A). However, the GRH biosynthesis genes were expressed at very low levels in the RA166-77 accession roots when compared with the HGRHA roots. In fact, the expression of these genes in the RA166-77 accession roots was similar to the expression levels in the LGRHA roots (Figure 8B and Appendix A). These results indicated that the GRH content was not dependent on the expression level of GRH biosynthesis genes in the radish roots.

## 3. Discussion

### 3.1. Abundant GRH Content and Functional Possibility for Radish Breeding

GSLs are sulfur-containing phytochemical compounds mainly found in the Brassicaceae. Radish is a root vegetable belonging to the Brassicaceae family, which has an abundant amount of a specific aliphatic GSL known glucoraphasatin (GRH) [18]. Previous research showed that GRH was the predominant GSL in the roots of eight radish varieties [16]. An analysis of the individual and total GSL contents from 44 *Raphanus* species revealed that GRH was the predominant GSL in both the leaves and roots of all the tested accessions [17]. The GSL profiling of the roots of 71 radish accessions showed that GRH was the predominant GSL in all accessions, constituting 95.2% of the total GSL content on average [18]. An analysis of the GSL content in Brassicaceae seeds showed that the GRH was only detected in two radish seeds, while it was not detected at all in broccoli (*B. oleracea* var. *italica*), rutabaga (*B. napus* var. *napobrassica*) and turnip (*B. rapa* ssp. *rapa*) seeds [31]. In the seven-day old sprouts of two radishes (Daikon and Sango) and two *Brassica* varieties (broccoli and Tuscan black kale), GRH was specifically detected in the two radish varieties [32]. Recently, GRH was detected in Chinese cabbage (*B. rapa* ssp. *pekinensis*), broccoli (*B. oleracea* var. *italica*) and choysum (*B. rapa* ssp. *chinensis*), but only at low levels [21,29].

In the present study, the GRH in roots accounted, on average, for 91.3% of the total GSL content in 59 radish accessions. Other GSLs, including GRA, GRE, GNA, GBN, GER, 4HGBS, 4MOGBS, NGBS and GBS, were at very low levels or non-detected in all radish accessions (Figure 1 and Appendix A). GRH contributes more to the production of isothiocyanates (ITCs) when compared with GRA, which is the major GSL found in broccoli [32]. Raphasatin (GRH-ITC) is a specific ITC derived from GRH in radish, whereas sulforaphane is derived from GRA in broccoli [22]. These ITCs are beneficial to human health, playing a role in preventing inflammation and carcinogenesis, and aiding in cardiovascular protection [18,33,34]. It is known that GRH-ITC possesses a selective cytotoxic/apoptotic activity on three human colon carcinoma cell lines [22].

GRH was significantly abundant and the predominant GSL in the roots of the radish accessions that were tested. The RA157-74 accession roots contained a significantly high GRH content; this HGRHA will be exploited in an expression study of GSL biosynthesis genes, as well as to improve a new variety of healthy resources.

### 3.2. Induction of GRH Biosynthesis by MYB29 Transcription Factor in Radish Roots

The aliphatic GSL biosynthesis process consists of side-chain elongation, core structure synthesis and side-chain modification in the Brassicaceae family. *BCAT4*, *BCAT3*, *MAM1*, *IPMI_LSU1*, *IPMI_SSU2, IPMDH1*, *BAT5, CYP79F1*, *CYP83A1*, *SUR1*, *UGT74B1*, *UGT74C1*, *SOT17*, *SOT18*, *MYB28* and *MYB29* genes are known to be the major genes involved in aliphatic GSL biosynthesis [6,11,12,35,36]. Furthermore, the *GRS1* gene has been reported to contribute to the accumulation of GRH in radish [29,30]. We measured the expression of these genes by RNA sequencing (Appendix A). In previous studies, radish mutants lacking GRH, *grs1-1* and *1-2* mutants, were grafted with a cultivar, and the majority of aliphatic GSLs in the radish roots were transported from the leaf to the root [30]. The *GRS1* gene, which is responsible for GRH production, is predominantly expressed in the leaf development stages (8 and 12 weeks old) of radish [29]. These studies indicate that the radish root was not the main site of GRH biosynthesis.

The genes involved in aliphatic GSL biosynthesis, including *GRS1,* were significantly expressed in the HGRHA roots when compared with all the tested accessions (Figure 4, Figure 5, Figure 6, Figure 7 and Figure 8). It was predicted that these genes would be induced by *MYB29* (*RSG00789*). However, the MYB29 transcription factor is not the dominant regulator, only playing a minor role in the induction of aliphatic GSL biosynthesis genes [25]. In Arabidopsis, a knockout mutant of MYB29 did not influence the expression of aliphatic GSL biosynthesis genes, whereas the MYB28 knockout mutants had significantly decreased levels or lacked these genes [25]. In this study, the expression of *MYB28* (*RSG53581*) was high in the IT119282-15 accession roots, but it had no significant effect on the expression of GRH biosynthesis genes. Meanwhile, one *MYB29* paralogue, *RSG00789,* was strongly amplified in the HGRHA roots (Figure 6, Figure 7 and Figure 8).

Several biotic elicitors, such as the phytohormones methyl jasmonate (MeJA), jasmonic acid and ethylene, increase the GSL content in the Brassicaceae family [31,37,38,39]. Aliphatic GSL biosynthesis has been known to be increased by MeJA in Arabidopsis. Under normal conditions, *MYB28* is more highly expressed than *MYB29*. MYB29 probably plays a supplementary role in response to MeJA signaling in Arabidopsis [40]. Our study revealed that some genetic factors influenced the expression of *RSG00789*, which probably led to the overexpression of GRH biosynthesis genes in HGRHA roots.

### 3.3. Expression Profiling of GRH Biosynthesis Genes in Radish Roots

In the side chain elongation step of aliphatic GSL synthesis, the precursor amino acid is deaminated to 2-oxo acid by BCATs. 2-oxo acid then undergoes a step involving three continuous transformations. Condensation with acetyl-CoA by the MAMs, isomerization by IPMIs and oxidative decarboxylation by IPMDH are known processes involved in the side chain elongation step in the chloroplast [28]. *BCAT4*, *MAM1*, *IPMI_LSU1*, *IPMI_SSU2* and *IPMDH1* genes were significantly differentially expressed in the HGRHA roots when compared with all tested accessions, but not with *BCAT3* (Figure 4B, Figure 6 and Figure 7). In Arabidopsis, BCAT3 has been shown to change the proportions of aliphatic GSLs in the terminal step of the side chain elongation process in response to amino acid biosynthesis [26]. Arabidopsis BCAT3 mutation did not decrease the total GSL content derived from methionine. The *BCAT3* gene mainly plays a role in leucine biosynthesis but can also act as a supporter of BCAT4 by transaminating methionine to 2-oxo acid in the side chain elongation step [26]. Expression of the *BCAT3* gene (*RSG45676*) is not necessary for GRH biosynthesis but seems to participate in leucine biosynthesis in radish roots [26]. All enzymes involved in the side chain elongation step are located in the chloroplast, except for BCAT4. Transporters are necessary for the import of 2-oxo acids into the chloroplast and for the export of chain-elongated amino acids to the cytosol. The *BAT5* gene encodes a bile acid transporter [28]. Both *BAT5* and *MYB29* are commonly induced by MeJA in Arabidopsis [41]. *MYB29* (*RSG00789*) and *BAT5* (*RSG43366*) were significantly expressed in the HGRHA roots when compared with all the tested accessions (Figure 7 and Appendix A).

During the core structure synthesis process, precursor amino acids are converted into aldoximes by CYP79F1, as seen in Figure 4. Next, the aldoxime is converted to S-alkyl-thiohydroximate by CYP83A1, and then the basic GSL skeleton is formed by the sequential activities of SUR1, UGT74B, UGT74C, SOT17 and SOT19 [6,42]. The two SOTs are responsible for the glucosylation of DS-GSLs [28]. Two genes, *SOT17* (*RSG03557*) and *SOT18* (*RSG07388*), were highly expressed in the HGRHA roots when compared with all the tested accessions (Figure 6 and Appendix A). The SOTs have different substrate specificities. The chain length of aliphatic DS-GSLs is the important element for substrate affinities of SOTs in Arabidopsis [28,43]. SOT17 and SOT18 prefer long-chain DS-GSLs derived from methionine, such as 7-Methylthioheptyl-GSL (7MTH) and 8-Methylthiooctyl-GSL (8MTO), but they are also active with short-chain aliphatic methionine-derived GSLs, such as 3-Methylthiopropyl-GSL (3MTP) and 4-methylthiobutyl-GSL (4MTB). SOT18 showed a broader substrate specificity than SOT17 in Arabidopsis [43]. GRH, the predominant GSL in radish, is a short-chain GSL derived from 4-methylthiobutyl (4MTB). The expression of *RSG03557* was significantly specific for 4MTB when compared with *RSG07388* in radish roots.

In the Brassicaceae family, FMO_GS-OXs_ are responsible for the modification of GSL side chains to produce methylsulfinyl types such as GRA, which are derived from methylthio types such as GER by oxygenation [29]. Seven *FMO_GS-OX_* genes have been reported in Arabidopsis [29]. We identified a total of six *FMO_GS-OX_* paralogous genes in radish. The expression of these genes was very low or not detected in the HGRHA roots, even though GRH is a methylsulfinyl-GSL (Figure 4 and Appendix A). Recently, it was reported that *GRS1* is directly responsible for GRH production [30], and we found that *GRS1* was highly expressed in the HGRHA roots when compared with all of the tested accessions (Figure 5).

In mature radishes, most genes involved in aliphatic GSL biosynthesis are found in the leaf. These genes are not expressed much in mature radish roots [11,29]. However, GSL biosynthesis genes were strongly expressed in the HGRHA roots. These results suggest that GRH can be significantly synthesized in the mature radish root under special conditions and that *MYB29* (*RSG00789*) plays an important role in this process.

Based on these results, we outlined a putative GRH biosynthesis process in HGRHA roots in a schematic diagram (Figure 9).

## 4. Materials and Methods

### 4.1. Plant Materials

A total of 59 radish (*R. sativus* L.) accessions were collected from the National Agrodiversity Center (http://genebank.rda.go.kr) (32 accessions of IT, Korea), the Leibniz Institute of Plant Genetics and Crop Plant Research (24 accessions of RA, Germany), and the National Agriculture and Food Research Organization (three accessions of JP, Japan). These accessions of *R. sativus* are referenced in Seo et al., 2018 [13]. For the analysis of the GSL content, seven seeds from each accession were sown in plastic pots, after which the seedlings were transferred into the field. After eight weeks, taproots from each mature radish were collected for an analysis of the GSL content and RNA-sequencing.

### 4.2. Determination of GSL Content

GSLs were extracted using the method described by Lee et al., 2018 [5]. A total of 100 mg of freeze-dried powder with 1.5 mL of 70% (*V*/*V*) MeOH was incubated in a water bath at 70 °C for 5 min. After centrifugation at 4 °C for 10 min (4000 rpm), the supernatants were harvested into a 15 mL tube. The supernatants were used as crude GSL extracts. These extracts were applied to DEAE-Sephadex A-25 (GE Healthcare, USA) in a minicolumn and were then desulfated using 75 μL of aryl sulfatase solution. Desulfo-glucosinolate (DS-GSL) was eluted with 1.5 mL of ultrapure water. The separation of DS-GSLs using a reversed phase Inertsil ODS-3 column (150 mm × 3.0 mm × 3 μm) was carried out with an E type cartridge guard column (10 mm × 2.0 mm × 5 μm) in an Agilent Technologies 1100 series HPLC system (Agilent Technologies, Germany) under the following conditions: the detection wavelength was 227 nm, column oven temperature was 40°C and flow rate was 0.2 mL/min. The solvent A (water) and solvent B (acetonitrile) consisted of the mobile phase. Elution of the DS-GSL samples was performed under the following gradients: 0 min, 100% A/0% B; 2 min, 100% A/0% B; 7 min, 90% A/10% B; 16 min, 69% A/31% B; 19 min, 69% A/31% B; 21 min, 100% A/0% B; and 27 min, 100% A/0% B. The classification of GSLs was dependent upon their HPLC retention time with our database. Individual GSLs were quantified using an external standard sinigrin (0.1 mg/mL for its desulfation).

### 4.3. Total RNA Isolation and RNA-Seq Analysis

The total RNAs were isolated from the roots using a Hybrid-R kit (GeneAll, Songpa-gu, Korea) according to the manufacturer’s instructions, and their quality and quantity were examined using a Bioanalyzer (Agilent Technologies, Santa Clara, CA, USA). About 2 μg of total RNAs were used to construct RNA-Seq libraries with an insert size of 300 bp, using an Illumina TruSeq RNA Sample Preparation Kit (Illumina, San Diego, CA, USA), according to the manufacturer’s instructions. Pooled libraries were sequenced using the Illumina HiSeq X platform with paired-end (PE) reads of 101 bp at Macrogen Co. (Seoul, Korea). Low-quality and duplicated reads and adapter sequences were removed using Trimmomatic (ver. 0.38) [44] with default parameters (removing a read with an average base quality below 20 and dropping a read less than 50 bases long). Trimmed high-quality RNA-Seq reads were mapped on the radish genome sequences of the *R. sativus* var. *hortensis* cv. Aokubi doubled haploid (DH) line [12] using HISAT2 (https://ccb.jhu.edu/software/hisat2/index.shtml) [45]. Then, RNA reads mapped on protein coding sequences (64,657 for var. *hortensis*) were counted using HTSeq-count (https://pypi.org/project/HTSeq) [46]. Fragments per Kilobase of transcript per Million mapped reads (FPKM) values were calculated using the number of RNA-seq reads mapped on protein coding sequences and were used for the expression profiling of genes.

### 4.4. Analysis of Differentially Expressed Genes Involved in GSL Biosynthesis

The bioconductor package DESeq (http://bioconductor.org/packages/release/bioc/html/DE-Seq.html) [47] was used to identify differentially expressed genes (DEGs) between samples with a low GSL content, medium GSL content and high GSL content, respectively. Genes showing over two-fold expression changes with an adjusted *p*-value of less than 0.05 were considered to be DEGs. A gene ontology (GO) enrichment analysis was performed for the DEGs using Fisher’s exact test with an adjusted *p*-value of 0.05 in BLAST2GO software (ver. 5.2, https://www.blast2go.com) and was then filtered by a *p*-value of 0.01, false discovery rate (FDR) of 0.01 and fold-change of selected genes to total genes > 2 for an analysis of GO terms. Information on GSL biosynthetic genes was obtained from previous studies [11,12], after which a GO analysis was carried out by functional annotation using nr BLAST, InterProScan and Araport11 database. The expression values (FPKM) of the genes were retrieved from the expression profiles of total gene sets and were used to generate heat maps using pheatmap in the R-studio.

### 4.5. Quantitative Real-Time PCR

A total of 1 µg RNA extracted from radish roots was synthesized using the RNA to cDNA EcoDry™ Premix (TaKaRa). Subsequently, 1 µL cDNA with 0.5 µL (10 pmol) of primers (Appendix A) and 10 µL Topreal qPCR premix, SYBR Green with low Rox (Enzynomix) was used for the quantitative real-time PCR (qRT-PCR). A 20 µL reaction mixture was subjected to amplification using a LightCycler^®^ 480 (Roche) under the following conditions. An initial denaturation at 95 °C for 10 min was followed by 45 cycles of denaturation at 95 °C for 10 s, annealing at 60 °C for 15 s and elongation at 72 °C for 15 s. The radish actin gene was also simultaneously amplified as a reference. The quantification of the gene expression was normalized to that of actin and was then calculated using the delta CT method.

## Figures and Tables

**Figure 1 ijms-21-05721-f001:**
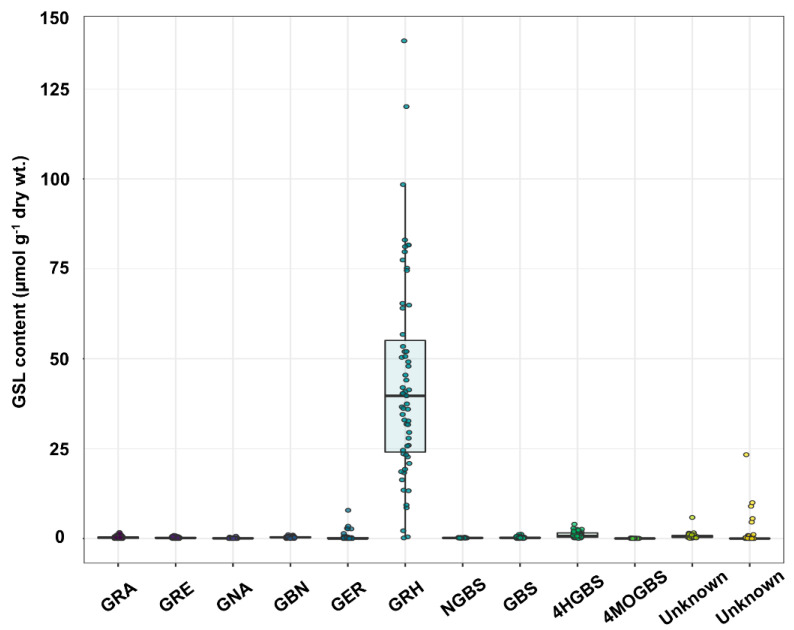
GSL profiling of a total of 59 radish accessions. Ten different GSLs, including glucoraphanin (GRA), glucoraphenin (GRE), gluconapin (GNA), glucobrassicanapin (GBN), glucoerucin (GER), glucoraphasatin (GRH), 4-hydroxyglucobrassicin (4HGBS), 4-methoxyglucobrassicin (4MOGBS), glucobrassicin (GBS) and neoglucobrassicin (NGBS) were identified in the radish roots.

**Figure 2 ijms-21-05721-f002:**
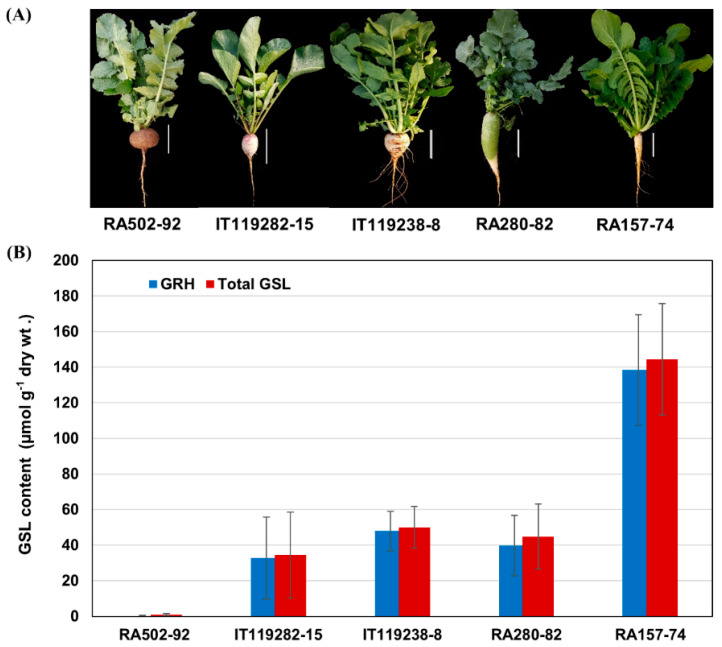
Phenotype images and GSL content in the roots of RA502-92, IT119282-15, RA280-82, IT119238-8 and RA157-74 accessions. (**A**) The phenotypes of the five radish accessions. Scale bars represent 10 cm (white bars). (**B**) Total GSL and GRH content. The blue and red columns represent GRH and total GSL, respectively. The error bars represent the SD (standard deviation) of three replicate experiments.

**Figure 3 ijms-21-05721-f003:**
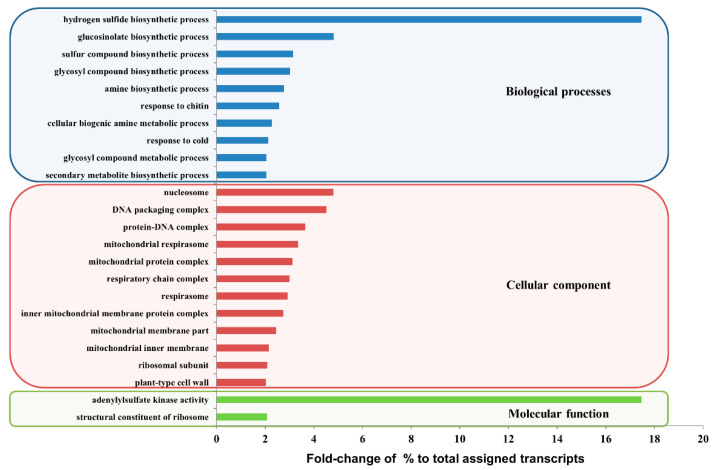
Functional classification of the gene ontology (GO) of DEGs between the HGRHA and LGRHA roots. Up-regulated genes in the HGRHA roots, when compared with the LGRHA roots, were divided into three main categories (biological processes, cellular component, and molecular function). The GSL biosynthetic process ranked second in the biological processes category. All GO terms were filtered by the following values; *p*-value < 0.01, false discovery rate < 0.01, fold-change of selected genes to total genes > 2. The *x*-axis indicates the fold-change as a% of total assigned transcripts. Detailed information is listed in Appendix A.

**Figure 4 ijms-21-05721-f004:**
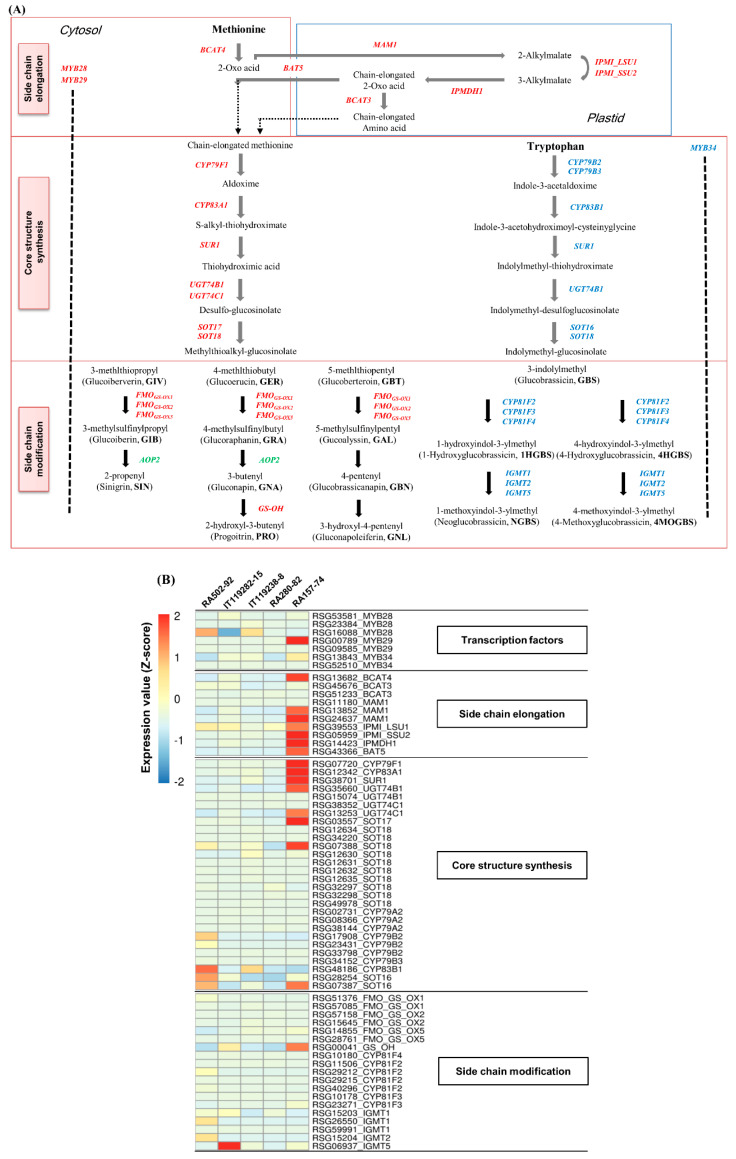
GSL biosynthesis in the Brassicaceae family. (**A**) Genes involved in aliphatic and indolic GSL biosynthesis are marked in red and blue, respectively. Undetected genes in the transcriptome of radish roots are noted in green. Dotted arrows in the side chain elongation step indicate the putative function of unknown enzymes. (**B**) Heat map analysis of genes involved in GSL biosynthesis in radish roots. Expression values (FPKM) of individual genes were normalized by Z-score and scaled per row (i.e., per gene) for the visualization of the expression peaks of genes among the accessions.

**Figure 5 ijms-21-05721-f005:**
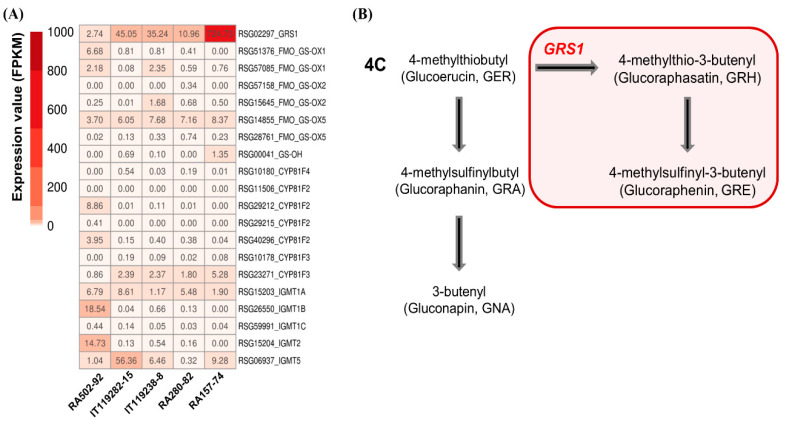
Heat map analysis of genes involved in side chain modification during GSL biosynthesis and the GRH modification step. (**A**) Heat map analysis and FPKM value of genes involved in side chain modification. (**B**) Side chain modification step in common 4-carbon aliphatic GSLs in the Brassicaceae family. The red box indicates the specific side chain modification for GRH production in radish. GRS1 encodes glucoraphasatin synthase 1.

**Figure 6 ijms-21-05721-f006:**
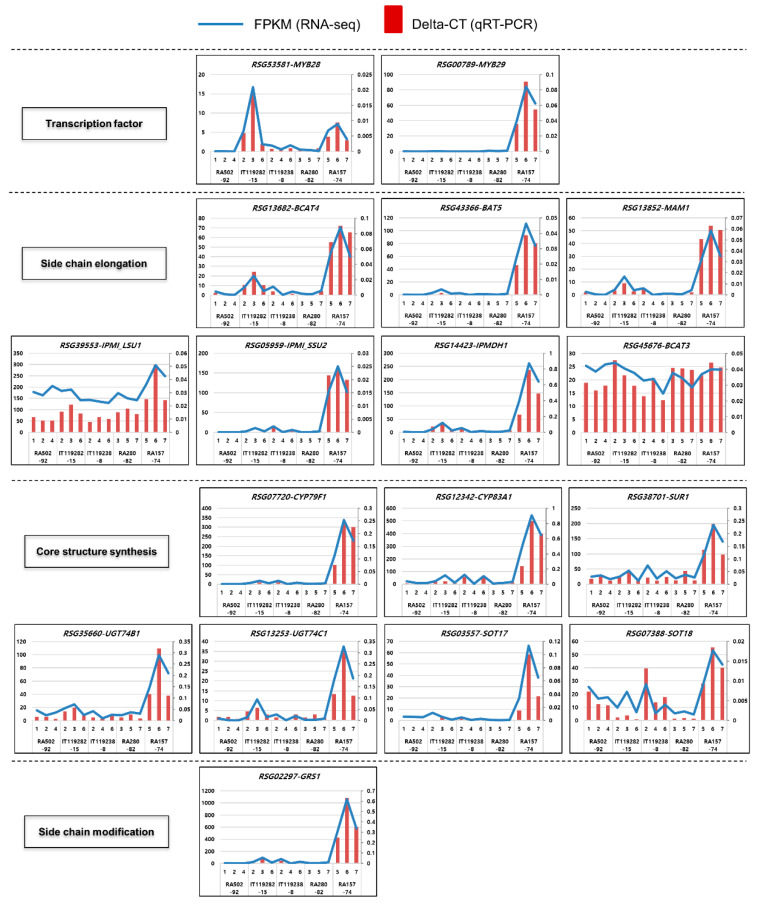
qRT-PCR examination of aliphatic GSL biosynthesis genes in radish roots. The GSL biosynthesis genes were analyzed by qRT-PCR, and the value was compared with the FPKM value. The blue lines represent the FPKM (left *y*-axis) data, and the red bars indicate the qRT-PCR data (right *y*-axis). The expression level of qRT-PCR was measured by the delta CT method.

**Figure 7 ijms-21-05721-f007:**
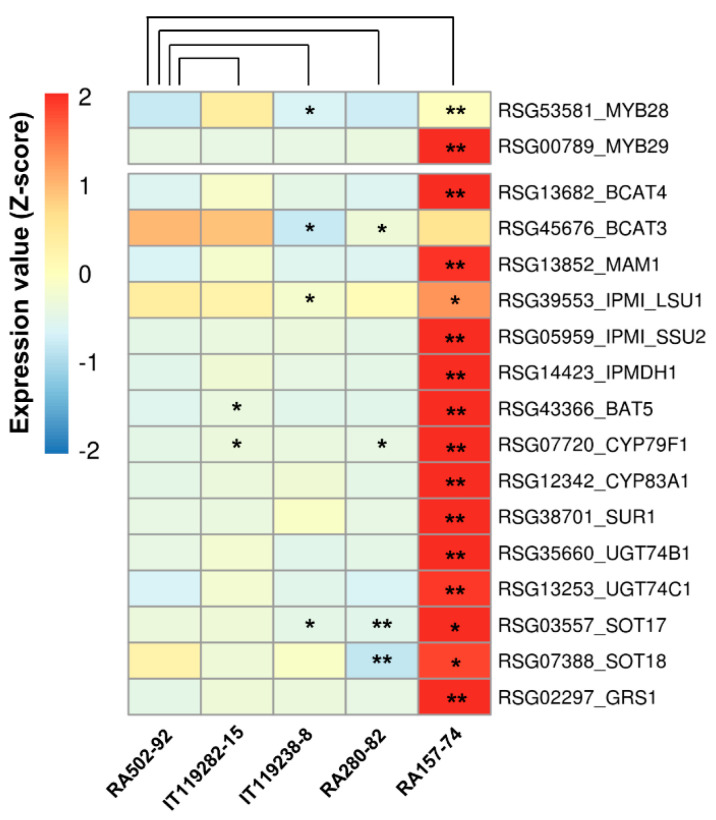
Analysis of DEGs involved in the GRH biosynthesis in radish roots. Expression values (FPKM) of individual genes were normalized by Z-score and scaled per row (i.e., per gene) for the visualization of the expression peaks of genes among the accessions. Significant differences are indicated as * *p* < 0.05 and ** *p* < 0.01 (Student’s *t* test).

**Figure 8 ijms-21-05721-f008:**
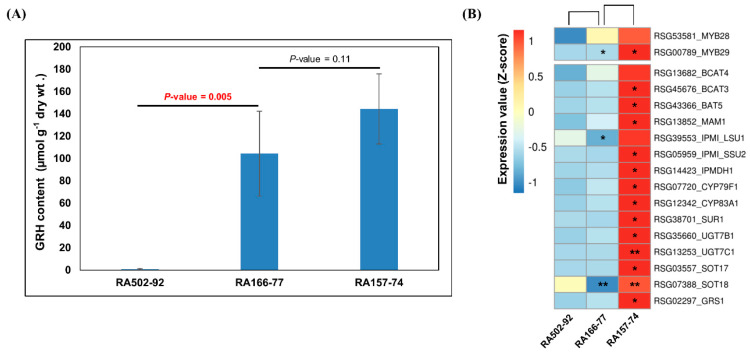
GRH content and analysis of DEGs involved in the GRH biosynthesis in radish roots. (**A**) Statistical analysis of GRH contents. (**B**) Heat map of DEGs. Expression values (qRT-PCR) of individual genes were normalized by Z-score and scaled per row (i.e., per gene) for the visualization of the expression peaks of genes among the accessions. Significant differences are indicated as * *p* < 0.05 and ** *p* < 0.01 (Student’s *t* test).

**Figure 9 ijms-21-05721-f009:**
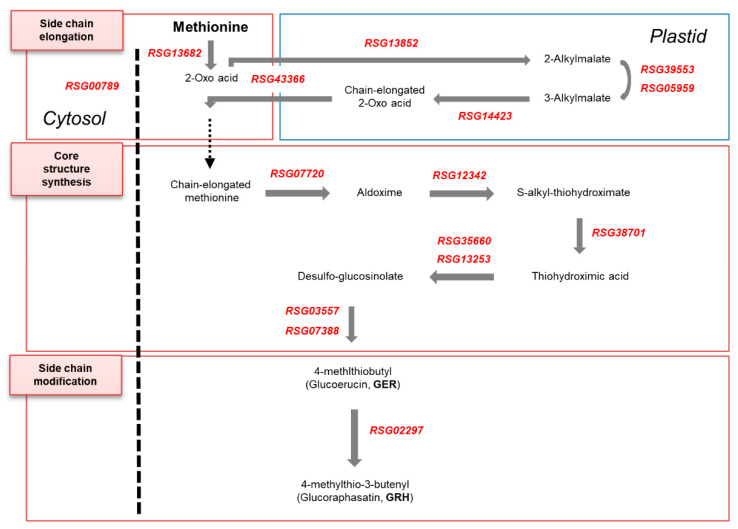
GRH biosynthetic pathway in radish roots. The dotted arrow in the side chain elongation step indicates the putative function of the unknown enzyme.

**Table 1 ijms-21-05721-t001:** Properties of GSLs found in radish roots.

Classification	Common Name	Abbreviation	Chemical Name	Molecular Formula
Aliphatic	Glucoraphanin	GRA	4-(Methylsulfinyl)butyl	C_12_H_23_NO_10_S_3_
	Glucoraphenin	GRE	4-Methylsulfinyl-3-butenyl	C_12_H_21_NO_10_S_3_
	Gluconapin	GNA	3-Butenyl	C_11_H_19_NO_9_S_2_
	Glucobrassicanapin	GBN	4-Pentenyl	C_12_H_21_NO_9_S_2_
	Glucoerucin	GER	4-(Methylthio)butyl	C_12_H_23_NO_9_S_3_
	Glucoraphasatin	GRH	4-Methylthio-3-butenyl	C_12_H_21_NO_9_S_3_
Indolic	4-Hydroxyglucobrassicin	4HGBS	4-Hydroxyindol-3-ylmethyl	C_16_H_20_N_2_O_10_S_2_
	4-Methoxyglucobrassicin	4MOGBS	4-Methoxyindol-3-ylmethyl	C_17_H_22_N_2_O_10_S_2_
	Neoglucobrassicin	NGBS	1-Methoxyindol-3-ylmethyl	C_17_H_22_N_2_O_10_S_2_
	Glucobrassicin	GBS	3-Indolylmethyl	C_16_H_20_N_2_O_9_S_2_

**Table 2 ijms-21-05721-t002:** GSL content in the roots of the five selected accessions.

Accessions	Origin	Species	Selfing Generation	GRA	GRE	GNA	GBN	GER	GRH	NGBS	GBS	4HGBS	4MOGBS	Unknown	Unknown	Total GSLs
RA502-92	Hungary	*Raphanus sativus* convar. *sativus*	S_2_	ND	ND	0.27 ± 0.05	0.04 ± 0.04	ND	0.17 ± 0.30	0.17 ± 0.04	0.01 ± 0.01	0.05 ± 0.06	0.08 ± 0.07	0.09 ± 0.15	ND	0.35 ± 0.05
IT119282-15	Lebanon	*Raphanus sativus* var. *sativus*	S_3_	0.08 ± 0.13	0.12 ± 0.12	ND	0.66 ± 0.39	ND	32.70 ± 23.09	0.19 ± 0.05	0.23 ± 0.21	0.16 ± 0.14	0.01 ± 0.02	0.27 ± 0.22	ND	62.35 ± 9.14
IT119238-8	Iran	*Raphanus sativus* var. *sativus*	S_2_	0.17 ± 0.29	0.39 ± 0.31	ND	0.46 ± 0.18	ND	47.92 ± 11.01	0.18 ± 0.02	0.20 ± 0.09	0.36 ± 0.23	0.07 ± 0.01	0.35 ± 0.10	ND	38.55 ± 5.43
RA280-82	Kazakhstan	*Raphanus sativus* convar. *sativus*	S_3_	0.74 ± 0.24	0.32 ± 0.14	0.23 ± 0.05	0.56 ± 0.26	ND	39.69 ± 16.99	0.17 ± 0.04	0.38 ± 0.19	2.06 ± 0.58	0.02 ± 0.04	0.68 ± 0.34	ND	60.12 ± 6.30
RA157-74	Portugal	*Raphanus sativus* convar. *sativus*	S_3_	0.85 ± 0.47	0.49 ± 0.37	0.22 ± 0.02	0.71 ± 0.22	ND	138.39 ± 31.11	0.19 ± 0.08	0.25 ± 0.11	1.68 ± 0.50	0.19 ± 0.04	1.51 ± 0.35	0.08 ± 0.13	122.46 ± 14.39

GSL content indicated in µmol·g^−1^ dry wt. Data presented as the mean ± SD of three individual roots. ND, not detected.

**Table 3 ijms-21-05721-t003:** Functional annotation for the *GRS1* gene (*RSG02297*) in a public database.

Gene ID	BLAST	
*RSG02297*	nr BLAST	PREDICTED: 1-aminocyclopropane-1-carboxylate oxidase homolog 5-like, BAW81934.1| GLUCORAPHASATIN SYNTHASE 1 (*Raphanus sativus*)
	InterProScan	IPR026992 (Pfam); Non-haem dioxygenase N-terminal domain,
		IPR005123 (Pfam); Oxoglutarate/iron-dependent dioxygenase
	Araport11	AT1G03410.2 | 2-oxoglutarate (2OG) and Fe (II)-dependent oxygenase superfamily protein | Chr1:844782-846463 REVERSE LENGTH = 361

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
