# Peer review of "Induction of Glucoraphasatin Biosynthesis Genes by MYB29 in Radish (Raphanus sativus L.) Roots"

_ijms, 2020, doi:10.3390/ijms21165721_

Round 1
Reviewer 1 Report
The authors found that in the roots of the high-GSL line, RA157-74, genes for the synthesis of aliphatic GSLs were highly expressed in the root. Furthermore, They suggested that one of the reasons for this was the expression of the transcription factor MYB29. The manuscript is technically sound and the individual data presented are compelling. A few suggested modifications to the tables and figures below.
Fig.2. The amount of glucosinolates is hard to see. Please consider whether three decimal digits are required.
Fig.5(B). The arrow of GRS1 should be written horizontally.
Fig.6. [recommend] It's better not to indicate the FPKM value with a line.
Author Response
Thanks to your revison work.
I attached the word file.
Sincierly yous, KJS

Reviewer 2 Report
Since glucoraphasatin synthase1 (GRS1) was found for the first time in 2017 by Japanese group(reference 31 in manuscript), there was little information on regulation of GRS1 expression and characterization of the enzyme in aliphatic glucosinolates biosynthesis in radish plant during their development. The present manuscript contains highly valuable information in that using comparative transcriptome analysis, RNA sequencing, and RT-PCR of the genes involved in aliphatic GSL biosynthesis, the author found MYB29 as a new candidate which may contribute to the biosynthesis of GSLs, including GRH in radish plant.
1. In the introduction between line 51 and 56, it would be advised that the importance of glucoraphasatin be emphasized in both biological and potential benefit as a health-promoting compound.
2. In Figure 1, it will be better to put the percent of total glucosinolates in each GSL on the top of the bar. In addition, it is advised to insert breakage in GRH data to have higher bar graph for the other GSLs to aid easy understanding.
3. The last sentence in 367-369, the authors claimed that MYB29 plays important role in the biosynthesis of GRH. It is suggested that the temporal relationship between aliphatic glucosinolates content and level of genes expression is necessary in order to address this question. For instance, in Table 2, it is recommended to include content of aliphatic glucosinolates besides GRH as being shown in Fig.1, and then compare results of HeatMap in Fig 7 and 8.
4. Since outline of GRH biosynthesis has been known in several reports, Fig 9. can be included in the Fig 4, then should be emphasized the uniqueness of radish aliphatic GSL biosynthesis.
5. Table 1, side chain should be molecular formula.
6. Some of the corrections be made:
Fig 5B and Fig 9: Glucoraphasatin should be 4-methylthio-3-butenyl and glucoraphanin be 4-methylsulfinyl-3- butenyl.
Line 287, sulforaphene should be sulforaphane othewise it provides wrong information.
Author Response

(The authors gave the same response as above.)

Reviewer 3 Report
In their manuscript, Kang et al. analysed glucoraphasatin (GRH) biosynthesis in radish. First, they analysed glucosinolate profiles of 59 radish accessions. From this data, they selected 5 accessions (with 3 biological replicates each), representing a wide range of GRH levels, for transcriptome analysis. Afterwards, differential gene expression analysis was conducted to identify genes related to GRH biosynthesis. Expression patterns derived by RNA-seq were also confirmed by qPCR.
While the methods and analyses conducted in this study are generally technically sound, the manuscript overall suffers from the weakness and lack of novelty of its key hypothesis, unfortunately. Major revisions will be required to make this report suitable for publication:
Major issues:
- the authors claim that MYB29 induces GRH biosynthesis. This is the key message of the paper, stated in the title, abstract, etc. However, the authors only show a correlation of MYB29 expression with other GRH biosynthesis genes, and no functional evidence at all. Previous reports (e.g. ref 26) have already examined this and related MYB transcription factors in Arabidopsis in much more detail, for example by gene knockout experiments. In addition to limited novelty, this claim is also not experimentally backed up.
- the manuscript contains several flaws in statistical analyses and data handling.
line 68: When analysing their 59 accessions, they tested "two or three biological repeats". This is unclear to me, how did they determine proper GRH levels from 2 replicates? At least 3 replicates should have been tested for each accession.
Figure 1: Plotting only the average GSL contents omits a lot of data. Please consider using a boxplot etc. for showing the variability of your data.
Figure 4B and 7: How did you convert the FPKM values into the relative changes from -2 to 2? Is this a log scale? Please describe how you transformed the data.
lines 18, 152, 237, 308, etc. "significant expression" / "significantly high". Please describe what you mean - did you perform any statistical analyses? If yes, provide details. Significant in comparison to what?
- line 15: "selected five, three and one accession roots [...] for conducting the comparative transcriptome analysis". This is highly misleading to me. I only see RNAseq analysis of 5 accessions and not 9?
- please deposit your RNAseq data in sequencing archives such as SRA and provide accession numbers
Minor issues:
- line 354: 4-meth[y]lthiobutyl-GSL
Author Response

(The authors gave the same response as above.)

Round 2
Reviewer 3 Report
The authors have sufficiently addressed all my comments - thank you. The manuscript is now ready for publication from my point of view.